# Examination of Short Supply Chains Based on Circular Economy and Sustainability Aspects

**Konrád Kiss [1,\*], Csaba Ruszkai [2] and Katalin Takács-György [3]**

[1]   Faculty of Economics and Social Sciences, Szent István University, H-2100 Gödöllő, Hungary
[2]   Innoregio Knowledge Centre, Eszterházy Károly University, H-3300 Eger, Hungary;
     ruszkai.csaba@uni-eszterhazy.hu
[3]   Keleti Faculty of Business and Management, Institute of Management and Organisation, Óbuda University,
     H-1084 Budapest, Hungary; takacsnegyorgy.katalin@kgk.uni-obuda.hu
\*   Correspondence: konrad.kiss@phd.uni-szie.hu

**Abstract:** The sustainability of global food chains and intense agricultural production has become questionable. At the same time, the consumers' interest in short supply chains (SSCs) and direct sales from producers has increased. SSCs are connected to sustainability by researchers. Their (supposed) positive sustainability attributes are based mostly on extensive production methods and short transport distances. However, from other points of view, the economic and environmental sustainability of the short chains is questionable. Our research aims to cast light on the SSCs' role in circular economy and sustainability. By deep literature review and content analysis, we determine the sustainability aspects of short (local) chains and their effects related to economy and environment. Short supply chains are connected most widely to circularity and sustainability by the subjects of environmental burden (transport, production method, emission), health, food quality, consumers' behavior, producer-consumer relationships, and local economy. According to our experience, these factors cannot be generalised across all kinds of short chains. Their circular economic and sustainability features are dependent on their spatial location, type, and individual attitudes of the involved consumers and producers.

**Keywords:** short supply chains; local food; food waste; environmental burden; consumer behavior; producers

---

## 1. Introduction

Generally, it can be said that agricultural food producers have to struggle for their (successful) operation—or even their survival—at low returns. The emphasis is on economic efficiency; for this reason, the ecosystem is under severe pressure [1]. Global food systems have a significant contribution to greenhouse gas emissions at all stages of the food chains: production, processing, marketing, sub-sale, home preparation and waste phases [2], and they can be one of the main causers of environmental harm like climate change, eutrophisation or loss of biodiversity [3]. In this way, the sustainability of traditional agri-food systems has been questioned over the last decades [4]. In developed countries, food consumption can contribute to greenhouse gas (GHG) emissions by up to 15–28% (based on national studies between 2007 and 2010) [5,6]. Increasing population; increased food demand, inadequate use, and distribution of food resources and high levels of food waste in the food systems are predicting the need for more sustainable practices [7]. Furthermore, consumers may be sceptical or distrustful of the intensive agricultural systems. The environmental and health effects of certain widely-used chemical substances are debated. For example, Toretta et al. (2018) [8] presented a scientific debate on the effects of glyphosate as the most sprayed and most used herbicide in the world.

Short food chains (SFSCs) have increasing popularity nowadays [9]; however, their role in the food trade of developed countries is limited [10]. They are unable to "replace" the global food systems in the lives of consumers [11,12]. They have the potential of supporting local and regional development and contributing to food quality for consumers [9] and job creation [13]. There are indications in the literature that these chains are capable of contributing to sustainability, due to their particular characteristics and small scale, although some aspects that are considered to be beneficial are also disputed by the experts' opinions [14].

Widespread dissemination of sustainability [15,16] and circular economy [17] is a priority of the European Union. Sustainable municipal solid waste management in a circular economy view has become an important goal in EU and non-EU countries [18]. Moreover, short supply chains (SSCs) are also supported in the EU budgetary period between 2014 and 2020 [19]. The objectives of the circular economy model overlap with the aims of sustainable development [20]. This overlap refers to the environmental and resource-saving nature of economic systems and their economic and social sustainability. In September 2015, the United Nations (UN) adopted an ambitious Agenda for 2030, setting 17 sustainable development goals for economic growth, social integration, and environmental protection. Goal 12 refers to "Responsible Production and Consumption" and relates to food loss and waste management [6]. The purpose of the subpoint 12.3 is: "By 2030, halve per capita global food waste at the retail and consumer levels and reduce food losses along production and supply chains, including post-harvest losses" [15] (p. 22). In the "Closing the Loop—An action plan for the Circular Economy" [21], there are five main priorities identified regarding circular economy, and "food waste" is one of them [22]. Worthy of mention is that sustainability and the Sustainable Development Goal 12 are related to other Goals like "Good Health and Wellbeing" (Goal 3) and "Life on Land" (Goal 15) [23].

The purpose of the current article is to systematize and consider the aspects whereby short trade chains can contribute to a circular and sustainable economic and environmental system. The sustainability of food chains includes three dimensions: economic, ecologic, and social [24–26], which can be complemented by personal health or well-being [25]. In connection with some of the objectives of the European Union circular economy package [17] and the Sustainable Development Framework [16], in our article we reviewed the following subjects: Food- and package-wastes in the (short) food chains, the environmental impact, economic and social characteristics and sustainability of the short supply chains, and short food chains. Within social characteristics, we specifically dealt with consumer welfare and health issues. Our main research question is that whether the belief in short supply chains is true that they are considered environmentally friendly compared to multi-actor retail chains and they can contribute to the development of the local economy and the well-being or satisfaction of consumers.

Our literature review was conducted by reviewing 128 sources (120 articles and 8 European Union professional materials). The most important source of our research was the Scopus database, where we searched for articles based on searching term "short food supply chain", started from the year 2011. We narrowed the search fields to agriculture, energy, environmental science, business, management, and economics. We sought to obtain information dealing with the relations of short supply chains, sustainability, and circular economy. We also used other databases such as EBSCO Discovery Service, ScienceDirect, AgeconSearch, Google Scholar, and searched for additional articles on the topics of circular economy and sustainability in general.

## 2. Literature Review

### 2.1. Conceptual Approach

In our article, on the basis of European Union subsidy policy (Regulation (EU) No 1305/2013), we consider a short supply chain (SSC), where producers sell their products to consumers directly, or through one intermediary [27]. Researchers dealing with this topic can find numerous alternative concepts, such as "short supply chain" (SSC), "short food supply chain" (SFSC) "alternative food

network" (AFN), "local food system" and so on. These terms generally refer to the spatial proximity of production and (final) sales and consumption, to local food products and a low number of participants in the chains. In our article, we made no distinction in content between such names; they were used alternately based on the different source works. (The spatial aspects of short supply chains' determination are not addressed in this article.) Direct sales and direct marketing have several types [12]. The most well-known are farm shops, farmers' markets, delivery of vegetable boxes by subscription, mail-orders, producer co-operatives, solidarity purchasing groups, and Community Supported Agriculture (CSA). The popularity of certain models varies from county to country.

It is widely believed that short supply chains (SSC) are suitable for the trade of high-quality products while promoting sustainability and efficiency [28]. The literature of this subject examines aspects like the reduction of food waste, eating healthier and more sustainable food, and ethical considerations. Participants of short food supply chains have increased knowledge and information on food and its origin [29]. However, many researchers are sceptical about the general optimism towards SFSC channels. Their supporters believe that local production is more sustainable than multi-actor retail chains, but this is less quantifiable. It would be dangerous to accept the disputed or disputable issues as absolute truths [30]. Further empirical research is needed to explore the sustainability impacts attributed to the certain alternative food chains [14].

The circular economy is one of the most popular research areas in the field of sustainability [31]. The concent of circular economy aims circular flows in the economy (opposed to the "linear flows" are dominant currently) [32]. It represents an economic model based on the recycling, reuse, repair, sharing, and leasing of existing materials and products [17]. The model of the circular economy can be interpreted in food chains regarding waste reduction (and minimization of surplus), food reuse, nutrient recycling, and the promotion of more varied and effective dietary patterns. It can affect the different levels of production and consumption [7].

In the case of new EU member states (joined in 2004), sustainability easily falls into the background due to the cost-efficiency of global food chains and their high level of influence [33]. The examination of "sustainability performance" of supply chains is still in an early stage, and it is definitely true in the case of short food chains [34].

Sustainability of food chains can be measured by the following indicators: Life cycle analysis (LCA), carbon accounting, material flow analyses, ecological foot-printing, food miles, Hazard Analysis and Critical Control Points (HACCP) studies, stakeholder dialogue and surveys and converting impacts into financial analysis [25,35]. In terms of environmental impact, the importance of the water allowance coefficient (WAC)—as an indicator based on water footprint—is also increased [36].

Examinations of local food systems' sustainability are made difficult by the fact that even consolidated approaches like a Life Cycle Assesment [37] cover only a small part of sustainability factors. Many aspects of sustainability are not yet fully measurable in a complex way. From the indicators mentioned above, our study focuses mostly on the analysis of food miles, because in this respect, the most striking difference between short chains is the shorter transport distance compared to the multi-actor chains.

## 2.2. Waste Originates from Food Chains

The amount of waste generated at various levels of food systems is huge, even by the most optimistic estimates. This means a very serious problem. According to FAO (Food and Agriculture Organization of the United Nations) data, nearly a third part of all food destined for human consumption is "lost" or wasted every year [38]. It is estimated that in the United States [39], or the European Union [7], this proportion can be as high as 30–50%. In terms of quantity, Buisman et al. (2019) [40] estimate the annual food waste in food chains in the EU at 89 million tonnes. According to the estimation presented by Corrado and Sala (2018), [41] the food waste means 194–389 kg per person per year at the global scale, and 158–298 kg per person per year at the European scale. (Worthy of mention is that according to the "Closing the loop- an EU action plan for the Circular Economy" [21], there is no

harmonised, reliable method to measure food waste in the EU. In this way, it is not surprising that the estimates are different. The "Food Waste Atlas" is a freely accessible online tool, where data about global food loss and waste from the food supply chain can be examined together [42].

An important element in achieving sustainable consumption and production is to reduce food waste [43]. Certain conventional treatments of food waste can lead to environmental, economic, and social problems. However, there are more sustainable or profitable management options. Reuse is possible, for example, in the chemical industry; consumers chemicals, acids, sugars can be synthesized from food waste, for instance. They can also serve as feedstock for biofuels and can be used in other ways [44]. However, the general, "linear" (non-circular) supply chain structures basically do not focus on reducing the negative aspects of production systems [26].

Composting, anaerobic digestion (AD) or innovative treatments as hydrothermal carbonization (HTC) are examples of food waste management. Anaerobic digestion seems to be one the most suitable solution for food waste treatment [45] and it has great potential for the disposal of the organic fraction (consequently food) of municipal solid waste [46]. This is a renewable energy source with low emissions [45]. Using products (e.g., fertilizers) created by hydrothermal processes like supercritical water oxidation (SCWO) or hydrothermal conversion (HTC) can contribute to large environmental savings [47].

Nayak and Bhushan (2019) [48] present the four basic approaches of valorization techniques of food wastes. The first is the conversion of food waste to generate biofuels; the second is to extract and (efficiently) recover various value-added components (e.g., proteins, pectins) from food waste; the third is "the conservation of food wastes via microbial activity to develop various bio-materials, in the form of bio-chemicals, bio-polymers, enzymes, single-cell protein, and bio-fertilisers" [48] (p. 364); and the fourth is to develop "effective absorbents from various bio-based food wastes for wastewater treatment" [48] (p. 365).

As food-waste can be generated at all stages of the food chain [39], efficient management for waste reduction requires the system approach of supply chains. The following methods of reducing food waste can be comprehensive and environmentally friendly: redistribution of unsold and surplus food, improving in-store promotion and stockpiling, reducing package sizes and stimulating or improving consumer perception and different consumer habits [49]. Waste reduction methods like food redistribution may have economic, infrastructural, and legal barriers [12]. For example, especially in rural areas, the network of food rescue organizations is often not tight enough to organize the transport of food surpluses from the point of origin to the food bank in an economically feasible way.

It should be noted that the primary approach of direct marketing is not based on waste reduction but on fostering community, preserving local food production, revitalizing the local economy, and protecting the environment. Furthermore, there are very few empirical studies on the impact of direct marketing on food waste generation. In the case of SSCs, the food chain is shorter and has fewer stations. The decrease in the number of intermediaries and traders—for various reasons—can significantly contribute to the reduction of food waste. In this way, losses caused by wholesale or retail may be decreased or zero. Such losses can be, for example, when they force producers to overproduce or refuse products that do not meet their standards [12].

According to a Swedish household survey, 20–25% of household food waste is related to packaging [6,50]. The sustainability of food packaging can be judged through three aspects: its direct impact on the environment, the quantity of food-losses and wastes related to packaging, and the circularity of packaging (reusability, recycling). On the other hand, food packaging also contributes to food protection; prevent it from being wasted early, becoming unfit for human consumption [51]. The role of food packaging varies in this way. Packaging minimization does not necessarily mean a complete solution to reduce the overall environmental footprint of foods [6,52]. In developing countries, much of the food loss is more likely to occur during the pre-consumer stages of the food chain, so the packaging can play a major role in protecting food. In the food chains of developed countries, waste is generated in a high proportion in the consumer phase (due to wasteful behavior) [7,38]. For this

reason, the behavior of individual households must also be considered to assess the environmental impact of packaging. It is significant how the consumers transport and store food, and how food waste and packaging waste are collected by them [3]. The objectives of the circular economy model include, for example, the use of recyclable packaging and the promotion of appropriate consumer attitudes, for example, by information, labeling [17], in order to prevent or reduce the generation of household waste [21]. In short supply chains, producers use less or zero packaging material. This is due to the nature of their economic and sales activities. The amount of sold products is less, and the purchasing process is less regulated than, let us say, in large chain-stores.

Food waste can also be linked to the appearance of the products. Consumers, or the chain-stores themselves, are reluctant to buy foods that, although just externally, differ from the optimum. It is possible to increase demand for such products through price discounts, or sustainability, quality and originality campaigns and positioning [53]. We have not found a reference to this in the literature, but in our experience, aesthetically defective vegetables and fruits are easier to sell in short chains, whereas they can not be placed on the shelves of large retail chains.

Consumer attitudes towards circular products are mostly still unexplored aspects of research. For example, it may be interesting to consider how many consumers are willing to return or dispose of used products [54]. The influence of social norms can provide opportunities for healthier, more sustainable diet patterns and habits [55]. At the same time, systems such as subscribed vegetable boxes can generate more waste than a common supermarket purchase; if consumers get some products they do not like or do not know how to prepare [6,12].

Customers' decisions are greatly influenced by where, how, and under what conditions their purchased products were made [56]. In multi-actor (global) chains, it is very difficult to trace the origin of food. As the supply chains become shorter (regarding the vertical phases), the traceability of products and related transactions are improved. Consumers can make more sustainable purchasing decisions if they have sufficient (usable) information [57,58]. Del Giudice et al. (2016) [56] examined the impact of food labeling on consumers' behavior. Waste-prevention-based labeling influenced consumers' willingness to pay positively. It was more affected by the carbon footprint frame of the reference (and not so much by the water footprint). According to this result, customers were interested in buying products that cause less environmental damage. In short food chains, the consumers and the producers get closer to each other; in this way, customers get accurate information on where the products originate [59].

Regarding food package, the large-scale plastic waste (including single-use plastic bags) is greatly responsible for the environmental burden. Generally, the shops provide a great amount of plastic bags for the customers. Offering eco-friendly reusable bags is a possible solution to reduce plastic waste [60]. The successfulness of these reusable bags is also dependent on the individual customers (or shops and sellers), who can be influenced by concerning campaigns. The authorities may also take actions, for example, banning, extra fees or taxes on plastic bags [60]. In terms of circular economy, the European Union set goals for 2030, in order to reduce plastic wastes, e.g., all plastic packages on the EU markets have to be either reusable or cost-effectively recyclable [61].

### 2.3. Economic Sustainability of Short Supply Chains

In terms of economic sustainability, in optimal cases, short trade chains can support local and regional development and contribute to the consumers' food quality [9] and job creation [13,62]. Experiences from several case studies show that SSC channels are mostly used by relatively small farms [63–65] (or are most profitable for them). They integrate supply chain functions in this way, but they also need to connect horizontally [66]. In many cases, producers' participation in SFSC is motivated by interdependence, self-employment [13], or by selling directly to the consumer at better prices, avoiding retail and wholesale trade [67]. In this way, they can receive a higher return on the value of the products [68]. Short chains have the opportunity to offer more value-added from producers [9]. Non-financial motivations of participating can be preserving tradition, establishing and

maintaining relationships with customers, protecting local, values, and environmental factors (such as sustainability and the natural or cultural environment) [69]. Limited local demand and seasonality of production mean drawbacks. According to Zhang et al. (2019) [70], in a wieder sense, SSCs can have a positive impact on local economic regeneration. Income generated by SSC participants may remain in the local economy [68]. In the case of the (orgainc) farms examined by Al Shamsi et al. (2018) [71], there was a strong correlation between the producers' preference of local markets and high performance in terms of sustainability.

The positives mentioned above come into being in the "ideal case" of course. Reduction of distances and the numbers of intermediaries may increase production earnings in short food chains, but in itself, it does not secure a long-term position in the food market [59]. The economic efficiency of the short supply chains depends on the particular situation and is highly dependent on the presence of solvent demand. There are consumers who are fully committed to local foods, but the numbers of such consumers are probably low. Other consumer groups are willing to give preference to local food, but only in a case of reasonable prices [72]. On a verbal level, there are many statements that certain forms of circular economy and short supply chains can play an important role in sustainable rural development, but on the other hand, these systems hardly exist in the newly acceded EU Member States (joined in 2004). In these new Member States, the relatively low willingness to pay for local products hampers the development opportunities for short supply chains [33]. Schupp (2016) [73] examined the location of producers' markets in the United States. According to his experiences, producers' markets are unlikely to appear in rural areas. As his results show, producers' markets affect almost exclusively the middle and upper classes, and they occur very rarely in areas with below-average socio-economic status. Also, in the American study of Low and Vogel (2011) [64], local food trade provided opportunity for economic development mostly in urban areas.

## 2.4. Connections between Short Chains, Sustainability, and Healthy Eating, Consumers' Well-Being

Unhealthy eating habits often develop during childhood and also persist in adulthood. Many young people do not have sufficient daily quantities of vegetables and fruits, as determined by the United States Department of Agriculture (USDA) [74].

Nutrition habits are based on complex decisions and have a significant impact on the environment and society [75]. From a consumers' perspective, several studies on SSC highlight aspects of freshness and "healthy eating" generally and specifically regarding fruit and vegetable trade (e.g., [76]). One of the main reasons for the preference of short chains is that consumers who choose SSCs perceive that small producers' wares are fresh, healthy, and of good quality. (e.g., [11]). Also, in the case of organic products' SSC-trade, healthy eating is the most motivating factor [77]. Ethical factors, such as traceability or environmental impact, can also have a significant influence on consumers' purchasing habits [78]. According to Leglise and Smolski (2017) [79], it is important for customers who prefer SFSCs to produce these goods with the best environmental practices, specific techniques, without agri-chemicals or industrial methods. In this way, product quality is also an attraction for customers [34]. At the same time, reliability, reputation, and respect of a (bio) producer may have a greater influence on consumers' behavior than the perceived bio-classifications on food [80]. In the traditional multi-actor chains, it is difficult to make good consumers' decisions regarding sustainability. Consumers are unaware of the entire food chain and its factors [7]. Consumers can be motivated or influenced by various campaigns to raise awareness of food waste and its environmental impact [53,81]. Del Giudice et al. (2016) [56] noted that by providing information on the carbon footprint of the given bread production, it was possible to motivate consumers to buy lower-carbon-footprint bread [53,56].

In the case of short chains, consumers are more aware of the place of foods' origin, the mode of production, and the identity of producers [70]. In the study of Giampietri et al. (2016) [82] where Italian students were surveyed, aspects like sustainability, and local development (as well as comfort) played a key role in short-chain shopping. Consumers who are spreading health awareness, expect healthy, fresh products. After harvesting, the quality of the products deteriorates continuously [83]. This is

very much related to the time between harvesting and getting to the table [55]. Stahlbrand (2015) [84] describes the food sector as a sector of "relentless deadlines". Food is perishable, and consumers demand "immediate service".

It is assumed that consumers attributing high value and good quality to producers' goods behave more consciously, and do not accumulate unnecessary surpluses that become waste later.

However, empirically, it is difficult to substantiate that local food would universally surpass non-local or imported food in terms of its impact on the environment or consumers' health. According to Edward-Jones (2010) [85], there is no known (or there was no known) publication examining the nutritional differences between local and non-local products or the health effects of a completely local diet. The positives attributed to SSC products can be attributed to the transport and storage characteristics that are different from conventional (multi-actor) sales channels. The quality and content of the products change differently compared to long delivery. Based on the study of Verraes et al. (2015) [86], SSC-products may have different microbiological quality parameters and different food safety aspects than food traded in conventional chains. According to their research on dairy products, SSCs are slightly more exposed to food safety concerns. This opinion is supported by the results of Jancsó et al. (2017) [87], who examined the quality of raw bovine milk sold directly to consumers in Hungary.

## 2.5. Environmental Aspects of Short Supply Chains

The idea of the circular economy can be interpreted in the food systems, as reducing losses, wastes, and avoidable environmental damage caused by the food chains [55].

The topic of sustainability has been gaining relevance in land use studies. The bibliometric analysis (of articles from the period 1988–2017) by Aznar-Sánchez et al. (2019, p. 13) [88] shows that research on sustainable development and land use "focuses on a new form of agrarian management, such as organic farming, permaculture, and multifunctional systems." The authors suggest that future lines should address the development of circular economic systems in agriculture.

Furthermore, extensive production methods may affect nutrition and (in this way) human health. For example, organic products of plant origin are grown without chemical-synthetic pesticides or without readily soluble mineral fertilisers or sewage sludge and waste compost. It is widely believed that organic foods are healthier than conventionally produced ones. (Absence of pesticide residues has great importance in this term.) However, the actual effect of agricultural techniques on nutrient composition is still not clear according to Gennaro and Qualia (2003) [89]. It is difficult to give a final answer due to uncontrollable factors (like rainfall and sunlight that have influence on nutrient content.) According to Popa et al. (2019) [90] (still) more research is needed to draw unwavering conclusions about the superiority of organic food (compared to conventional ones.) Relatively more environmentally sustainable production methods may be associated with SSCs, resulting in less input use, including pesticides, synthetic fertilizers, animal feed, water, and energy [10]. Applying a shorter supply chain can, in financial terms, facilitate the application of more sustainable practices. Such practices may include feeding the farm-animals by local feed or grazing, using organic, or biodynamic cultivation methods, or using on-farm production of renewable energy [68].

The lower negative environmental impact attributed to short supply chains (e.g., [9]) can be explained by the reduction of the food-miles (distances between the place of production and consumption) due to lower $CO_2$ emissions or noise pollution [67].

However, in itself, the food mile as an indicator is not sufficient to asses the environmental impacts of food chains [14,91]. It is not enough to measure transport emissions because it is difficult to assign certain kilometers to particular foods. Furthermore, different modes of transport, equipment, and different types of fuel also should be taken into account [25]. Some authors equate shorter transportation with less energy use, while others consider that short chains basically have poor energy performance.

Many consumers seek to reduce harmful environmental effects by consuming locally produced foods [92]. However, the benefits of "eating locally" and energy use are being challenged in several

studies [93], and for the right judgment, the effectiveness of SSC distribution and the distances travelled by consumers for purchasing, must be examined. According to Weber és Matthews (2008) [94], the "buying local" behavior of an average American household could reduce greenhouse gas emissions proportionally only by 4–5%, since the majority of gas emissions occur during the food production phase.

From a consumers' perspective, the more the consumers have to drive for purchasing, the greater their environmental impact and $CO_2$ emissions [34]. Local food means an opportunity for sustainability if production, distribution, and consumers' shopping trips are sufficiently energy- and cost-effective [57]. Mancini et al. (2018) [34] suggest that buying in a specialized dairy shop (investigated by them) is less environmentally effective than going to a larger grocery store, where a wide variety of foods can be found. On the other hand, one of the drawbacks of SFSC-s in terms of logistics is the less concentrated transport, which also results in lower efficiency (small freights, that are time and money demanding, especially for longer distances, and to less populated areas with specific delivery conditions) [95]. This suggests that energy- and cost-effective mass delivery systems can be even more sustainable than local production and distribution [6,57]. However, sustainability cannot be measured with a single indicator (travelled distance, or greenhouse gas emission). Energy efficiency is not the only measure of sustainability [93].

Industrially produced meat consumption in Western nutrition is a critical factor for sustainability in food consumption [7]. Van Huis and Oonincx (2017) [96] state that due to the growing population, growing consumer demands, and limited land-areas, there is an increasing need to replace traditional meat products. Changing over to a more plant-based diet is a possible solution [7]. Small farmers selling in SSC can play a role in this process. A very important product category for short supply chain trade is the purchase of vegetables and fruits from producers. Vegetarianism, as a consumer choice, is often associated with its positive environmental effects. A vegetarian meal has a less environmental impact than a pork-based meal (with about 40% lower emission) [56]. In the case of livestock rearers, regional (local) feed supply can also have a positive environmental impact, but it can also significantly increase the consumers' price of meat and dairy products. The socio-ecological impact of price increases can be significant; on the one hand, it can contribute to the change of dietary habits, for example by switching to plant-based foods; however low-income consumers may not be able to effectively change their buying habits [97].

It is difficult to draw a comprehensive conclusion on the sustainability of SSCs because the farms involved and the production methods they use are very different. Truly sustainable food systems should have a low environmental impact and, according to Al Shasmi et al. (2018) [71], organic production is one of the best ways to achieve this. However, even SSC-trade and organic farming are not automatically considered environmentally friendly [30], nor can it be generalized that conventional supply chain farming systems would in all cases be environmentally more intensive than SSC-oriented ones [98]. Sustainability and product quality performance of SFSC-s are closely linked to the local context and market situation in which they operate [28]. Organic farming and integrated farming are often described as they can reduce the environmental impact, but for the production of various vegetables (such as salads and leaves), there is just a little scientific evidence on the relative environmental impact of such alternative production methods [99].

Local food systems using organic methods increase worldwide, but little is known about their carbon footprint. Vitali et al. (2018) [100] examined the production of greenhouse gases in short supply chains related to the marketing and sale of organic beef. They found that farm activities and home consumption had the greatest global warming potential in the product path. As a conclusion, the environmental impact of SSC transport was considered to be relatively low, compared to production and consumption.

Based on "arbitrary rules" [85], it can be said that in the case of seasonal fresh products, such as fruits and vegetables, the carbon footprint is lower, or at least comparable to non-local fruits and vegetables. The short travelled distance and the avoidance of intermediary actors and quality preserving processes mean shorter shelf-life and freshness [28]. But, for example, if fresh products are stored and consumed

out of season, the above-mentioned "arbitrary rules" may fail [85]. According to Frankowska et al. (2019) [101], green vegetables imported to England produced in unheated greenhouses have a lower environmental impact than vegetables produced locally in heated greenhouses in England (despite the transport). Operations during the processing phase like freezing, pasteurisation, baking, use of added materials (e.g., oil), also result in high(er) environmental impacts. Because of agro-ecological and socio-ecological differences, it is not certain that the environmental impact of local food is lower, for example, if a given production site is more suitable for producing a certain type of food, compared to other (closer) areas [102]. Because of these facts, in agreement with the words of Depperman et al. (2019) [97], one has to be very careful about statements that call local food equal to sustainable food. Consuming local food is not always an environmentally beneficial option. It also should be noted that many products cannot be raised or produced in local systems because of climatic conditions [103]. In this way, consumers have to purchase them in conventional trade chains, regardless of sustainability considerations.

According to Gatimbu et al. (2018) [104], the relationship between environmental and economic performance is not clear. Both positive, negative, and insignificant examples can be found in the literature. According to their [104] research with small-scale tea factories in Kenya, environmental efficiency reduced the economic efficiency of business. According to Lehtinen (2011) [25], cost-effectiveness is not necessarily opposed to sustainability, and local food systems are not necessarily sustainable, but there are several facts that support the view that local foods can be more sustainable than other alternatives.

### 2.6. Social Sustainability of Short Supply Chains

Regarding the sustainability of supply chains, research on social aspects is still underrepresented, and this fact offers further research opportunities for the future [105]. Examining societal aspect may be recommended directly in the field of Circular Economy too [106]. According to Taghikhah et al. (2019) [107], it is impossible to talk about sustainability without extending the supply chain to consumers' behavior itself and its impact on system performance. In their work, they show how a supply chain can increase its socio-environmental and economic performance by motivating consumers towards green consumption, and how consumers motivate producers (and suppliers) to change the way they operate in this regard. Consumers' campaigns [81] may be able to reduce consumers' waste production by highlighting harmful environmental impacts.

Production ways and methods can greatly influence consumers' decisions [56]. Supporting local producers can be an important motivating factor for consumers' participation in SSCs (besides their various attitudes towards food quality) [77,108]. Producers with strong consumer relationships can be greatly supported by the community [109]. The sustainable operation of SSCs strongly depends on the (producer and consumer) community that operates it. The success of farmers' work depends on the support of the community [110]. The long-term viability of SSC-channels such as CSA (Community Supported Agriculture) is highly dependent on customer satisfaction [111] because if producers establish long-term relationships with consumers, CSA can operate cost-effectively and optimally [11]. The social and environmental side of farming can also be a motivation factor for consumers. Promoting and sustaining other people's well-being is in line with the basic goals of short supply chains [112]. Even antipathy to the dominant consumer culture can motivate customers to buy in SSCs [113]. The visibility of food production and its natural and seasonal limitations may encourage customers to a sparing and responsible handling of food [12].

In general, determining the market price of new "green products" related to the circular economy is an important optimization problem. Substitute—and possibly cheaper—products on the market, make it difficult to develop optimal pricing and advertising strategies [114]. Customers generally have a positive attitude towards the locality of production, but this does not mean in itself that they are able and willing to pay premium prices for local products [30]. Local food is usually more expensive than conventional chain products due to low production volume and high (specific) transport costs [25].

In the case of premium-priced products, consumer willingness to pay is a major issue. D'amico et al. (2014) [115] investigated consumer habits in Italy, in the directly sold wine market. According to their results, prices did not have a decisive role in the selection of local products. Based on the results of Carpio and Isengildina-Massa (2009) [116], it was found that the willingness to pay was higher among responders who attributed a higher quality to local products (than to products from outside South Carolina, where the research happened). Consumers were willing to pay an average surplus of 27% for local products. However, it should be noted that the willingness to pay for local products may vary in space and time. Results from other studies may draw attention to a lower willingness to pay. It is worthy to emphasise the possible demand-stimulating effects of tourism. Local-, agro- or gastro tourism may have positive effects on local community and economy. (Local) food has an important role in (gastro)tourism, and according to an older research on American tourists [117,118], up to 25% of total tourist expenditure is accounted for by food. Agrotourism can play also an important role in protecting and preserving the environment [119].

On the topic of local communities, it is worth mentioning that according to Bavec et al. (2017) [120], the current literature on SSCs has not yet paid sufficient attention to the economical organisation of SSCs. They were not studied extensively from a business perspective [121]. The importance of trust and community awareness also comes to the fore in the organisation of short chains, because according to Van Oers et al. (2018) [65], they are essential for a high level of acceptance of organisational activities (e.g., in the cases of CSA-s). Trust between producers and consumers is based on the personal relationship of the participants [66], and their relationships have a mutual role [122] in community building. Loyalty and trust can contribute to the progressive development of SFSC-s [123]. They can create community bonds, but not under all circumstances [109]. Consumers of SFSC-s may also require that (local) products be associated with a local (cultural) identity [79,124]. Demartini et al. (2017) [30] also drew attention to the possible drawbacks of producer-buyer relationships, that direct contacts with consumers do not necessarily lead to higher profits or "fair" transactions. A profiteering farmer may exploit the consumers' confidence.

## 3. Materials and Methods

Very significant literary works have been created before in the term of circular economy, for example, the bibliometric network and survey analysis of Türkeli et al. (2018) [106], or the bibliometric analysis of 743 articles made by Ruiz-Real et al. (2018) [125] Sustainability and short supply chains are also widely researched topics. In this article, we relied on our experiences we gained from the articles of a literature review.

To systematize the information material, we selected aspects, expressions, and factors from the literature that were most decisive for the content of our article, in a brainstorming manner way. These terms or topics were examined in the reviewed articles on short supply chains. Fifty-three terms were collected in this manner (Appendix A). The purpose of this collection was to select factors related to the circular economy and sustainability aspect of short supply chains in terms of environmental, economic, social and consumer welfare (based on Lehtinen (2011) [25]), in a comprehensive way. All of these terms are connected to each other in a broad sense, to a certain extent. We endeavored to explore the more important relations based on the above-mentioned terms.

The 55 aspects we selected that are closely related to those "50 most common words" were collected by Tseng et al. (2019) [126], from articles on the subject of green supply chain management. According to our subjective judgment, our list covers approximately 70% of the 50 most common words in the content, while the word-for-word or close-to-word ratio is approx. 38%. The differences come from the fact that short supply chains differ in some characteristics from conventional trading chains, so some features, such as "management" or "technology" were not used in our analysis, while others (such as "trust", "employment") were used.

Takács and Takács-György (2019) [127] presented a list of the most mentioned terms of English language articles published in the "Annals of the Polish Association of Agricultural and Agribusiness

Economists" between 2009 and 2019 (It is one of the most important Polish scientific journals in the field of agricultural economics). From the 17 terms, eight occur in our list in an almost literal match, and 14 overlap in content.

## 4. Results

The 55 sustainability, circular economy aspects we collected were grouped according to four conceptual classes. They relate to the categories identified by Lehtinen (2012) [25], which cover the sustainability aspects of food chains. These are the "ecological", "economic", "social", and "consumer" dimensions. Then, we searched for relationships between the four dimensions formed in this way. This is how "Environmentally Conscious, Sustainable Production" and "Lifestyle; good community and healthy eating" dimensions arose (Figure 1) (A list of the identified aspects and the system of their connection can be found in the Appendices A–C, and in Figure A1).

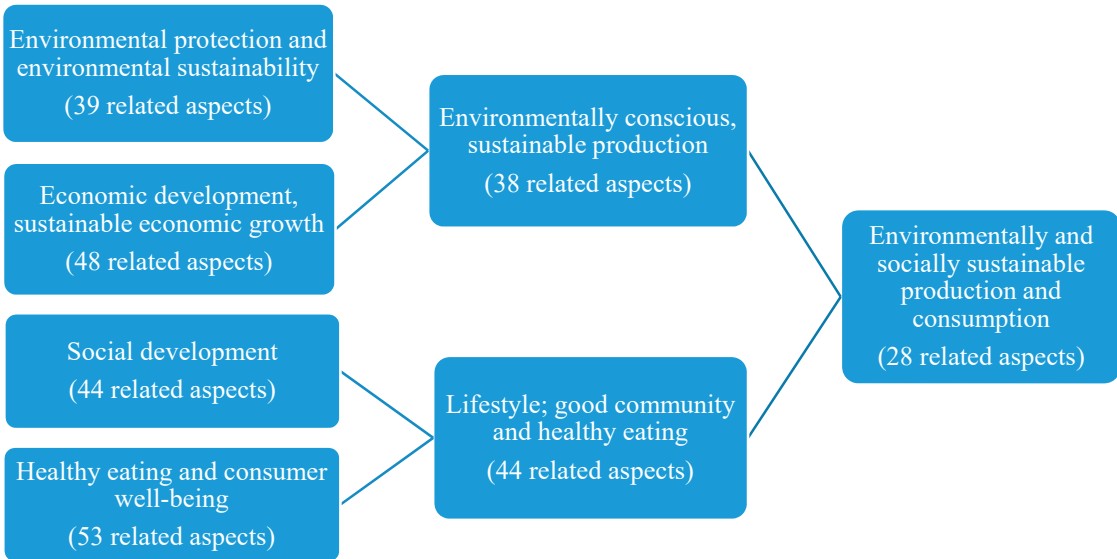

**Figure 1.** Grouping of the aspects of short supply chains related to sustainability and the circular economy (based on 55 concepts/aspects). Source: own editing, connected to Lehtinen (2012) [25], Tseng et al. (2019) [126] and "Appendices A–C".

Finally, we identified the general dimension which is related to both of the environmental, economic, social and consumer dimensions: "Environmentally and socially sustainable production and consumption". Among the concepts underlying the grouping, the following were comprehensively related to the "Environmentally and socially sustainable production and consumption": Carbon (-emission, -foodprint); circular economy; consumer (purchasing); cooperation; cost (producers'); education (producer and/or consumer); environment; environment friendly production; health; marketing/advertisement; package/packaging; policymakers/government; pollution; producer; price (consumers') rural; rural development; social/social embeddedness; sustainability; tourism/tourism destination; urban; waste; wellbeing (Appendix C).

From a sustainability perspective, these conceptual classes imply that people's well-being and (physical and mental) health is closely linked to the state, cleanness or pollution of their environment, home and to the quality of the food they consume, and their relationship with their community. The basic aim of the circular economic model is to use resources sparingly and considerately and to reduce the environmental burden in this way. Its successful operation requires the right attitude of producers, and shifting consumers' food purchasing habits towards sustainability, for example, by favoring low carbon footprint or food mile distance products, with a conscious behavior to avoid food waste and reduce waste generation.

By this means, the circular economy (waste reduction) and sustainability aspects of short supply chains form a close link with the environment, economy, and society. In Figure 2, we systematized our experiences, creating an "ideal, model-like" system where all of the presented aspects contribute positively to the goals of sustainability and the circular economy. It is "ideal" and "model-like" because—as we presented in the "Literature review", these aspects may also have their lacks and downsides, and their positive impacts cannot always be realized. It can be said that they are dependent on the given situations, for example, when the production methods used by small producers are not environmentally friendly, or if SSC-logistics is not efficient enough, or consumers' willingness to pay is low, or their behavior is not environmentally conscious (in greater detail, see the Literature Review chapter).

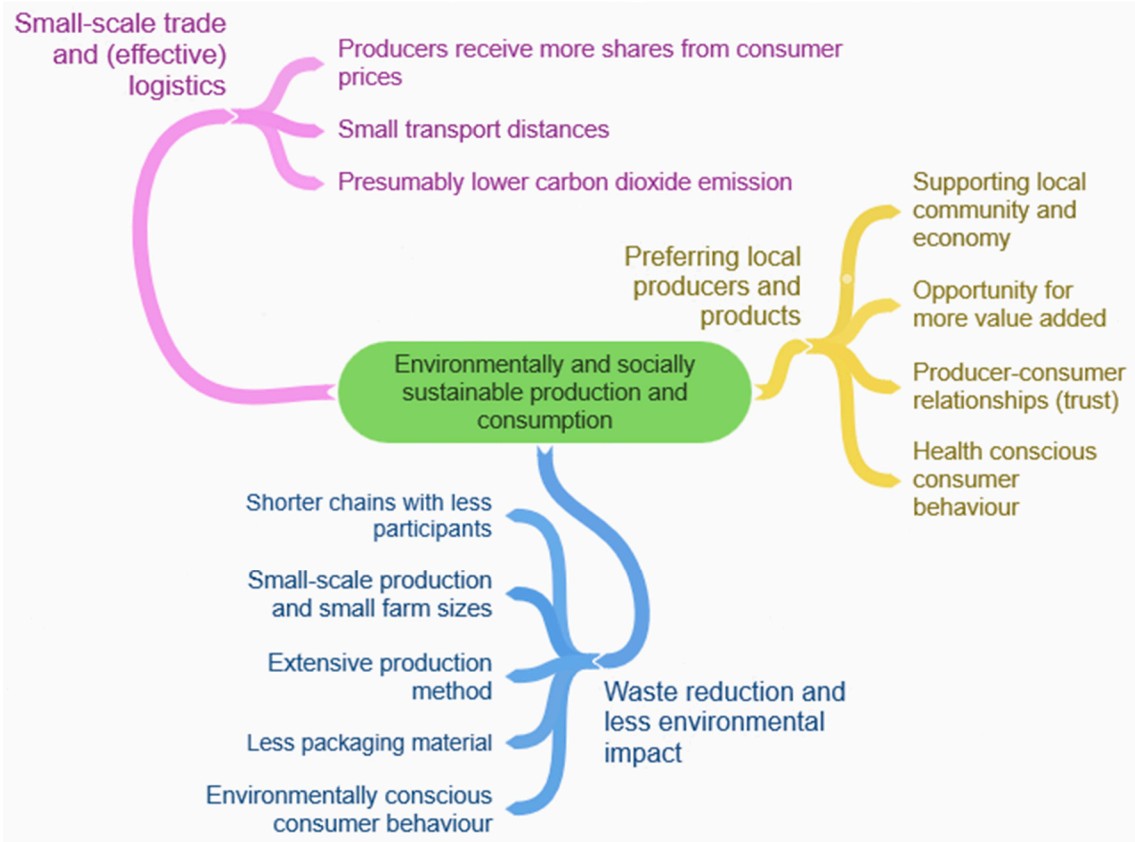

**Figure 2.** Possible positive effects of short supply chains on the circular economy and sustainability goals. Source: Own editing (used "coggle.it"), based on the literature review

## 5. Discussion and Conclusions

In our article, we reviewed the sustainability and circular economy aspects of short supply chains (SSCs) through a literature review.

The sustainability of food chains is linked to the dimensions of the environment, economy, society, and consumers' welfare [25]. We collected 55 concepts or factors that well describe the sustainability and circular economy implications of short supply chains along these four dimensions. On their base, the cross-section of the four dimensions is the "environmentally and socially sustainable production and consumption". This term indicates that, in our experience, supply chains generally can be brought into connection with sustainable production and consumption by the aspects of health, well-being, community, producers and consumer behavior, reduced waste and pollutant emission. Furthermore, the organisation and efficiency of short chains can be fundamentally affected by governmental support or regulatory policies. The effective operation of circular economic aspects requires the supporting behavior of producers and individual consumers.

The principles of the circular economy regarding food chains include minimising waste and surplus, reusing food, nutrient recycling, and promoting more varied and effective dietary patterns [5]. Although the concept of short supply chains is not based on waste reduction, in our experience, they can contribute to the prevention of food waste, and in this way, to the objectives of the circular economy. The trade of fresh products, with shorter shelf-lives, moderate packaging usage, flexible package sizes, and possibly more conscious customer behavior may contribute to the waste reduction. However, it should be mentioned that, as the role of SSCs in modern trade is very limited, these aspects have a less important role in large-scale waste reduction.

Reduced carbon emissions from short transport distances is an important fact for assessing the environmental impact of SSCs. Furthermore, food goes through fewer processing steps, with less or zero packaging, and the small-sized producers are likely to use extensive production methods. However, these findings are depending on the given situation: it is not regular that small producers always use extensive production methods and SSC transport may be less (environmentally) efficient due to its possible deconcentration (with numerous small freights [95], and greater distances travelled by customers.) Besides, the distance of transport and food miles—as indicators, in themselves—are not sufficient to assess the environmental impacts of food chains [91].

It is undisputed that there are many potentials for sustainability in short supply chains—provided that they meet the appropriate economic, environmental, and social conditions. However, following Born and Purcell [128], we agree that "local traps" should be avoided, which means local systems should not be automatically declared as "good practices", based solely on proximity. As Depperman et al. (2019) [97] suggested, one has to be very careful with statements that call "local food" equal to "sustainable food".

Finally, we need to mention the limitations of our research. We have endeavoured to collect a sufficient amount and quality of the literature as a sample, but it is more than likely that there is information that our research does not cover. We studied English-language journal articles, and this excludes the presentation of case studies and experiences from non-English journals. Assessing the sustainability of short supply chains and their role in a circular economy can be a more complex and multi-faceted task, to which our article sought to contribute.

**Author Contributions:** Conceptualization, K.K., C.R. and K.T.-G.; methodology, K.K.; software, K.K.; validation, K.K., C.R. and K.T.-G.; formal analysis, K.K.; investigation, K.K.; writing–original draft preparation, K.K.; writing–review and editing, K.K. and K.T.-G.; visualization, K.K.; supervision, K.T.-G.; project administration, C.R.; funding acquisition, C.R.

**Funding:** This research was funded by the project entitled: "EFOP-3.6.2-16-2017-00001 Complex rural economic development and sustainability research, development of the service network in the Carpathian Basin." (In original, Hungarian language: "EFOP-3.6.2-16-2017-00001 Komplex vidékgazdasági és fenntarthatósági fejlesztések kutatása, szolgáltatási hálózatának kidolgozása a Kárpát-medencében.")

**Acknowledgments:** The Authors wish to express their gratitude to the Szent István University for providing access to the necessary databases (e.g., Scopus). The Authors also thank their colleges and friends from the Szent István University and Eszterházy Károly University for their advice and encouragement contributing to this study.

**Conflicts of Interest:** The authors declare no conflict of interest.

## Appendix A

"Aspects describe the relation of short supply chains and sustainability and circular economy"

Accessibility; bio/organic; carbon (-emission); child; circular economy; consumer; cooperation; cost (producers'); delivery; education; employment; energy consumption; environment; environment friendly production; fairness; fair trade; family; food safety; food security; food quality; food-mile; handmade products (small amount); health; income; marketing/advertisment; nutrition/nutritional value; package/packaging; policymaker/governemnt; pollution; poverty; price; process/processing; producer; producer-consumer relationship; rural development; rural; social/social embeddeddnes;

supplying with food; sustainability; traceability; tourism/tourism destination; transport; trust; urban; waste; wellbeing; zero-kilometres (-distance -products).

*Appendix A.1 Environmental Protection and Environmental Sustainability*

Bio/organic carbon (-emission); circular economy; consumer; cooperation;cost (producers'); delivery; education; energy consumption; environment; environment friendly production; food-mile; handmade products (small amount); health; marketing/advertisment; package/packaging; policymaker/governemn; pollution; price; process/processing; producer; rural development; rural; social/social embeddeddnes; sustainability; traceability; tourism/tourism destination; transport; urban; waste; wellbeing; zero-kilometres (-distance -products).

*Appendix A.2 Economic Development, Sustainable Economic Growth*

Accessibility; bio/organic; carbon (-emission); circular economy; consumer; cooperation; cost (producers'); delivery; education;employment; energy consumption; environment; environment friendly production; fairness; fair trade; food quality; food-mile; health; income; marketing/advertisment; package/packaging; policymaker/governemnt; pollution; poverty; price; process/processing; producer; producer-consumer relationship; rural development; rural; social/social embeddeddnes; supplying with food; sustainability; traceability; tourism/tourism destination; transport; trust; urban; waste; wellbeing; zero-kilometres (-distance -products).

*Appendix A.3 Social Development*

Carbon (-emission); child; circular economy; consumer; cooperation; cost (producers'); education; employment; environment; environment friendly production; fairness; fair trade; family;food safety; food security; food quality; handmade products (small amount); health; income; marketing/advertisment; nutrition/nutritional value; package/packaging; policymaker/governemnt; pollution; poverty; price; producer; producer-consumer relationship; rural development; rural; social/social embeddeddnes; supplying with food; sustainability; tourism/tourism destination; trust; urban; waste; wellbeing.

*Appendix A.4 Healthy Eating and Consumer Well-Being*

Accessibility; bio/organic; carbon (-emission); child; circular economy; consumer; cooperation; cost (producers'); delivery; education;employment;environment; environment friendly production; fairness; fair trade; family; food safety; food security; food quality; food-mile; handmade products (small amount); health; income; marketing/advertisment; nutrition/nutritional value; package/packaging; policymaker/governemnt; pollution; poverty; price; process/processing; producer; producer-consumer relationship; rural development; rural;social/social embeddeddnes; supplying with food; sustainability; traceability; tourism/tourism destination; trust; urban; waste; wellbeing; zero-kilometres (-distance -products).

**Appendix B**

*Appendix B.1 Environmentally Conscious, Sustainable Production*

Bio/organic; carbon (-emission); circular economy; consumer; cooperation; cost (producers'); delivery; education; energy consumption; environment; environment friendly production; food-mile; health; marketing/advertisment; package/packaging; policymaker/governemnt; pollution; price; process/processing; producer; rural development; rural; social/social embeddeddnes; sustainability; traceability; tourism/tourism destination; transport; urban; waste; wellbeing; zero-kilometres (-distance -products).

*Appendix B.2 Lifestyle; Good Community and Healthy Eating*

Carbon (-emission); child; circular economy; consumer; cooperation; cost (producers'); education; employment; environment; environment friendly production; fairness; fair trade; family; food safety; food security; food quality; handmade products (small amount); health; income; marketing/advertisment; nutrition/nutritional value; package/packaging; policymaker/governemnt; pollution; poverty; price; producer; producer-consumer relationship; rural development; rural; social/social embeddeddnes; supplying with food; sustainability; tourism/tourism destination; trust; urban; waste; wellbeing.

## Appendix C

*Appendix C.1 Environmentally and Socially Sustainable Production and Consumption*

Carbon (-emission); circular economy; consumer; cooperation; cost (producers'); education; environment; environment friendly production; health; marketing/advertisment; package/packaging; policymaker/governemnt; pollution; price; producer; rural development; rural; social/social embeddeddnes; sustainability; tourism/tourism destination; urban; waste; wellbeing.

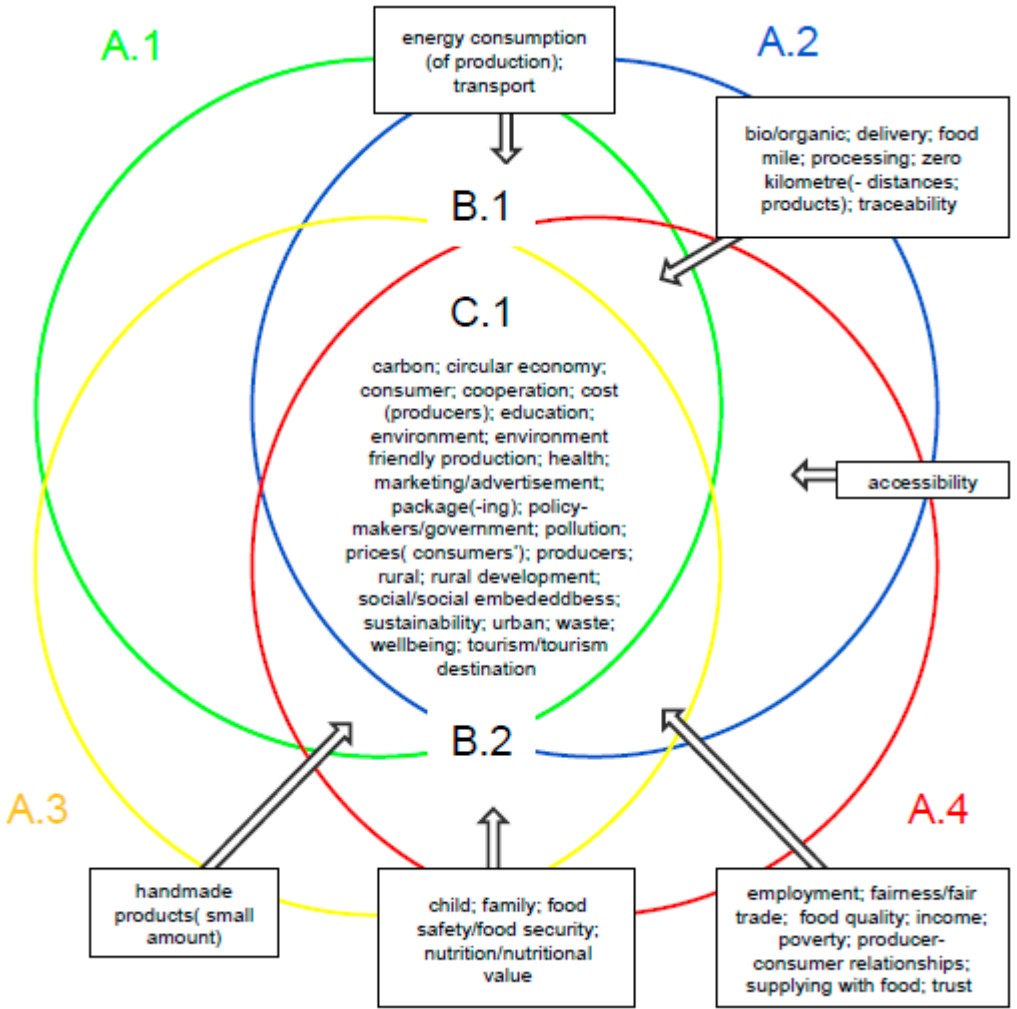

**Figure A1.** "Aspects describe the relation of short supply chains and sustainability and circular economy, and they possible*way of connections". Source: Own editing, (used "gliffy.com"). All of these terms are connected to each other to a certain extent, but this system focuses on supply chains, and reflects the opinions of the authors.

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
