# Peer review of "Examination of Short Supply Chains Based on Circular Economy and Sustainability Aspects"

_resources, doi:10.3390/resources8040161_

Round 1

Reviewer 1 Report

the paper presents an interesting review. However some modifications must be made before its acceptance.

See the attached PDF file with indications 

Author Response

Dear Ms. or Mr. Reviewer!

Thank you for your praises, remarks and advices on the improvements of our article.

We made all of the corrections you requested. The changes and new parts were remarked by red colour in the text. (A comment can be found on p 4.)

We used the new sources and references you offered (l. pp. 1-4; and 10) and even more. Reference list and its numbering were also changed.

We created new paranthese on the "plastic free" aim  of the EU (p. 5.); and we aspired to make one on the "regarding food treatment in a sustainable and circular economy view in order to decrease the environmental and heath impact." (p. 7.) As we experienced, we found the most important information on "land use" and "production method" regarding to this question. You also can see further information about the sustainability and circular economy aspects of food chains in the article, for example in the "Conceptual approach" (p.3.)

We made a rephrase on the sentence about "landfilling and incineration" (p.4.) We created a new Figure, describes the system of the aspects of our Method (Figure 3.; p 15.). Based on the new experiences, we also made changes in the list and relations of these aspects. 

We made corrections in the English.

Please, take a view on our changes, and we are waiting for your new feedbacks.

Yours sincerely:

Konrád Kiss

Reviewer 2 Report

The article raises a topic of interest for the Academy, short supply chains based on circular economy and sustainability aspects. The most positive aspects of this work are the following:

The introduction clearly presents the objectives of the work and the literature review is very complete, with a large number of references, most of them quite current.

As for the analysis of results and conclusions are correct, providing interesting reflections and implications for management.

Aspects that should be improved:

Review and include some of the current bibliometric analyzes performed on circular economy, such as:

Ruiz-Real, J.L.; Uribe-Toril, J.; De Pablo Valenciano, J.; Gázquez-Abad, J.C. Worldwide Research on Circular Economy and Environment: A Bibliometric Analysis. Int. J. Environ. Res. Public Health 2018, 15, 2699.

Turkeli, S., Kemp, R., Huang, B., Bleischwitz, R., & McDowall, W. (2018). Circular economy scientific knowledge in the European Union and China: A bibliometric, network and survey analysis (2006-2016). Journal of Cleaner Production, 197, 1244-1261.

The methodology used is explained very briefly. It would be convenient to detail it in more detail in the corresponding section.

Author Response

Dear Ms. or Mr. Reviewer!

Thank you for your praises, remarks and advices on the improvements of our article.

We made the corrections as you and your co-reviewer requested. The changes and new parts were marked by red colour in the text. 

We examined and added the references you adviced (p. 10.).

We aspired to extend the Material and Methods part.

We made a new Figure (F.3 in p 15.) presenting the relation of our used aspects. 

According to our new experiences we made improvements in the list of the aspects and their connections.

Please, take a view on our changes, and we are waiting for your new feedbacks.

Yours sincerely:

Konrád Kiss
